# Exploring Physician and Patient Perspectives on Expectations and Role Models Towards Chronic Pain Treatment in General Practice: A Qualitative Cross-Sectional Study

**DOI:** 10.3390/healthcare13020187

**Published:** 2025-01-18

**Authors:** Dominik Dupont, Sabrina Brinkmöller, Sarina Carter, Michel Wensing, Cornelia Straßner, Peter Engeser, Regina Poß-Doering

**Affiliations:** 1Department of Primary Care and Health Services Research, University Hospital Heidelberg, University Heidelberg, 69120 Heidelberg, Germany; dominik.dupont@med.uni-tuebingen.de (D.D.); sabrina.brinkmoeller@med.uni-heidelberg.de (S.B.); michel.wensing@med.uni-heidelberg.de (M.W.); cornelia.strassner@med.uni-heidelberg.de (C.S.); peter.engeser@med.uni-heidelberg.de (P.E.); 2Institute of Health Sciences, University Hospital, 72076 Tübingen, Germany; 3Clinic for Palliative Medicine, University Hospital Heidelberg, 69120 Heidelberg, Germany; sarina.carter@med.uni-heidelberg.de

**Keywords:** chronic pain, general practice, quality of care

## Abstract

**Background and Objective:** Approximately 7.4% of the German population matched the criteria of impeding, chronic non-cancer-related pain in 2014. Guidelines emphasize the importance of a holistic treatment approach. The project RELIEF aims to develop and evaluate a multifaceted case-management intervention to foster the holistic management of chronic pain in general practice. To inform intervention development, this study explored expectations and perceived role models of general practitioners (GP) and patients regarding chronic non-cancer-related pain management in general practice with a focus on patient expectations of GPs and themselves, as well as GP expectations of patients and their anticipation of patient’s expectations. **Methods:** Data were collected via semi-structured guide-based interviews with general practitioners and patients. Pseudonymized verbatim transcripts were analyzed using an inductive–deductive approach with a structuring qualitative content analysis. The Theoretical Domains Framework served as an analytical framework to explore behavioral aspects associated with expectations and role perceptions. **Results:**
*n* = 25 interviews were analyzed (GP: *n* = 10, patient: *n* = 15). Findings indicate that patients considered themselves as the main actor in the therapy process yet expected guidance and care coordination from their GP. The essential role GPs play in pain management was emphasized. Role models indicated by GPs and some patients represent a care ideal, which was also reflected in discussed expectations. GPs anticipated that patients would place high relevance on pharmacological options. Patients highlighted their preference of non-pharmacological and alternative treatment options. **Conclusions:** Our findings demonstrate the importance of holistic, individually tailored chronic pain management in general practice. Systematic, multifaceted case management, as planned in the RELIEF project, may contribute to high-quality primary care for affected individuals.

## 1. Introduction

In Germany, about one in five patients in general practices report pain that lasts longer than three months or is recurring and thus is chronic by definition [1]. Similarly, a systematic review and meta-analysis indicated a prevalence of chronic widespread pain of up to 24% in Europe, Asia and America [2]. It can be assumed that the healthcare needs of chronically ill individuals in general will increase with rising life expectancy and, therefore, chronic care will gain importance in the future [3]. In 2014 already, a total of 7.4% of the German population matched the criteria of impeding, chronic non-cancer-related pain (CNCP) [4] However, so far, inconsistent data on the prevalence of chronic pain in the German adult population have been reported [5]. A distinction can be made between two types of CNCP: (1) chronic primary pain, which cannot be attributed to diagnosable tissue damage, e.g., fibromyalgia or psychosomatic pain disorders, and (2) chronic secondary pain, which can be attributed to an underlying disease or injury [6]. CNCP does not only influence the life of affected individuals and their relatives, but is also associated with severe impairment of everyday coping, social isolation and reductions in health-related quality of life [7,8]. Hence, psychological comorbidities such as post-traumatic stress disorder, depression or anxiety are often associated with CNCP [9]. From an economic perspective, CNCP increases health services utilization, sick leave and early retirement [10]. In Germany, pain-related disabilities were associated with a 4.5-fold increase in physician visits and a 6-fold increase in sick leave [7]. This results in a relevant financial impact on health insurance and therefore on society [10].

Guidelines emphasize a holistic treatment approach that considers the biopsychosocial model, the significance of non-pharmacological measures and a rational pharmacological therapy [11,12]. Although it can result in medication overuse and a higher prevalence of adverse effects, pharmacotherapy is the most applied treatment option worldwide and in Germany [8,13]. Alternative treatment options include physiotherapy and activity and patient education, as well as psychological approaches [11,12]. In a German telephone survey, 72% of respondents with CNCP stated that they used over-the-counter non-steroidal anti-inflammatory drugs (NSAID), and only 10% of the patients were in specialized pain treatment [8], with a first consultation after 3–6 years of suffering from the condition [14]. In an international context, individuals with chronic pain in Germany make the least use of specialized pain therapist care [8].

General practitioners (GPs) play an essential role in the detection and treatment of CNCP in Germany, since they often have a longstanding relationship with their patients and are aware of their social, medical and personal circumstances. As a primary access point to the health system in Germany, GPs prescribe medication and write referrals to other medical specialists, as well as further therapeutic interventions such as physiotherapy or psychological therapy [15]. Although these are the best conditions for CNCP treatment in general practices, challenges exist within this care setting: applying recommended assessment and monitoring instruments in daily practice can be challenging due to limitations of time or knowledge, and a preference for verbal GP–patient communication [16]. Also, GPs might be hesitant to include emotional and spiritual aspects in their consultations [17].

Several qualitative studies, and one mixed-methods study, have indicated that patients with CNCP want their care providers to emotionally support them, take them seriously, provide holistic and individually tailored care, and consider illness and identity contexts [18,19,20]. None of these studies considered the GP perspective or were conducted in the German healthcare context. Various studies investigated the effectiveness of singular interventions for CNCP in the primary care setting [21,22], yet only a small number of studies evaluated the implementation of a multifaceted CNCP care program [23,24]. The project RELIEF (resource-oriented case management to implement recommendations for patients with chronic pain and frequent use of analgesics in general practices) aims to develop and evaluate a multifaceted case-management intervention to foster the holistic management of CNCP of various etiologies in general practice. Feasibility will be assessed in a before–after pilot study and subsequently, the case management will be tested in a confirmatory cluster-randomized controlled trial. This present study’s objective was to explore expectations and the perceived role models of GPs and patients regarding CNCP management in general practice, with a particular focus on patient expectations of GPs and themselves, GPs expectations of patients and their anticipation of patient’s expectations, to inform intervention development [25] and contribute to the field.

## 2. Materials and Methods

### 2.1. Study Design and Context

The RELIEF project was guided by recommendations on the development and evaluation of complex interventions [26]. In the initial phase, extensive literature reviews were conducted to provide an overview of the research field prior to data collection [15]. Based on the study team’s prior knowledge and practical expertise (experienced in medical and nursing pain management; co-authors of corresponding medical guidelines), and the knowledge gained during this process, a stakeholder survey and qualitative studies were conducted to inform intervention development. The RELIEF intervention will address five key components: software-supported pain assessment, scheduled structured appointments, e-learning on CNCP for GPs and medical assistants, educational material for patients, and a toolbox comprising evidence-based relevant information and links about regional resources such as outpatient and inpatient treatment services, as well as specialized psychological institutions in the federal state of Baden-Württemberg [25].

Based on generated qualitative data and a thematic literature research, this exploratory, cross-sectional qualitative study investigated GP and patient perspectives regarding expectations and perceived role models referring to chronic pain management in general practice.

All methods applied in this study were carried out in accordance with the Declaration of Helsinki, German legal regulations and the Guidelines for Safeguarding Good Research Practice of the German Research Foundation [27]. This study was approved by the Ethics Committee of the Medical Faculty of Heidelberg University (Reference-Nr.: 087/2023; 1 March 2023).

### 2.2. Study Population

Based on the experiences of the leading researchers of the RELIEF project, as well as the internal study protocol, a sample size of 40 participants was considered appropriate to reach thematic saturation. The inclusion criteria for patients and GPs were age (18 years and above), the ability to give informed consent and fluent mastery of German. GPs had to be practicing, fully trained and specialized in general practice, internal medicine or currently undergo specialist training. The patient inclusion criterion was to suffer from a CNCP condition. Patients with chronic pain resulting from oncological diseases or in palliative care were excluded.

### 2.3. Recruitment and Sampling

A purposive sampling strategy was followed to recruit patients as well as GPs [28]. It was intended that a heterogeneous sample of patients would be obtained with regard to age, gender, residential area (urban/rural) in the Rhine-Neckar region in Baden-Württemberg and duration of chronic pain. To reflect heterogeneity, general practices in 71 randomly selected urban and rural communities in the state of Baden-Württemberg, Germany, were contacted via their publicly available e-mail address (*n* = 900 practices) and invited to participate in an interview.

Patients were recruited for interview participation via the paper-based survey questionnaire that was mailed to a total of *n* = 3200 adult residents in six randomly selected municipalities in the Rhine-Neckar district of Baden-Württemberg. The questionnaire provided a contact form to express interest in participating in an interview and sharing perspectives on CNCP treatment in general practice. Addresses were obtained from resident registers. Since older people are increasingly affected by CNCP [4], the survey was initially sent to *n* = 1200 persons over the age of 70. In a second recruitment wave, *n* = 2000 residents between 18 and 69 years old were contacted to include perspectives of younger individuals.

In addition, the invitation was disseminated to self-help groups by e-mail via the project partner Gesundheitstreffpunkt Mannheim, an umbrella network located in Mannheim, Baden-Württemberg that hosts and connects self-help groups for various diseases. Furthermore, the invitation was distributed on social networks via the accounts of Heidelberg University Hospital. After expressing interest in participating in an interview, patients were sent an information leaflet and a consent form addressing participant confidentiality, followed by an informed consent discussion conducted via telephone to describe the study objectives and interview process.

### 2.4. Data Collection

In a structured approach, the project team designed semi-structured interview guides (Appendix A) that comprised four chronological steps: (1) the open collection of potentially relevant questions; (2) review, selection and structuring; (3) the sorting of remaining questions; and (4) establishing a logical order for the questions conducive to conversation [29]. The interview guide for GPs was piloted with a GP using the think-aloud method [30]. The patient interview guide was piloted in the first interview with a participating patient. The wording in both guides was slightly adapted after piloting and the coherence of both interview guides was verified. In the patient interview guide, medical terminology was adapted to increase the clarity of questions. Minor changes to the GP interview guide included the shortening of wording and adding prompts for optional in-depth questions. Interview duration was expected to be 30–45 min. The interview guides were complemented with a sociodemographic questionnaire.

All interviews were conducted in 2023 by two junior researchers with a background in health services research, pediatric nursing and pain management. A reimbursement of €50 was offered to all participants. Participants could choose between in-person or telephone interview. All interviews were digitally audio recorded and initially transcribed verbatim with the AI-assisted software NoScribe (Version: 03-2023). Transcripts were then checked for plausibility by the first author. Collected data from the interviews and sociodemographic questionnaires were pseudonymized and stored on secure servers of the Department of Primary Care and Health Services Research, University Hospital Heidelberg. Due to phone connectivity issues, one interview was recorded and subsequently transcribed in several parts, and then combined in one transcript. All interviews were conducted for exploratory purposes and covered several focal points. Written informed consent was obtained from all participants prior to the interviews.

### 2.5. Data Analysis

Prior to data analysis in this present study, all transcripts were screened regarding relevance for the defined research objective and a subset of the generated data was used for data analysis. The pseudonymized transcripts were first analyzed using an inductive–deductive approach with a structuring qualitative content analysis [31]. To manage, organize and analyze the data, the software MAXQDA 2022 was applied. The process of a structuring qualitative content analysis comprises seven steps: (1) initiatory familiarization with the transcripts; (2) the mostly deductive development of the main categories in accordance to the research question; (3) first step of coding with the main categories; (4) the inductive development of subcategories; (5) coding the data with subcategories; (6) simple and complex analyses; and (7) preparing the results for publication and dissemination [31]. Following step five, the Theoretical Domains Framework of behavior change to investigate implementation problems (TDF) [32] was applied as an analytical framework to explore behavioral aspects associated with expectations and role perceptions of GPs and patients. To characterize behavioral and organizational factors in healthcare settings, the TDF describes 14 theoretical domains, including several subconstructs contributing to the definition of the domain. It is based on 33 established theories to describe and analyze behavior, with a focus on behavior change in health contexts [32]. Following the inductive approach, main- and subcategories were deductively assigned to relevant TDF domains. The final system of categories was inductively completed by themes identified de novo from the data.

For the analytical frame of this study, the characterization of the terms, expectations and role models was based on sociological, literature-based definitions [33]: Expectations were considered to be based on past experiences and socialization, were future-oriented and contained assumptions and hopes about a situation or the behavior of another person. From a sociological perspective, the idea of expectations was closely associated with the concept of role theory. Role models were therefore based on references to the past as well as the resulting expectations and served as orientation within a social system to assign predictable actions. Roles included behaviors, values and expectations [33].

A quantitative analysis of the sociodemographic characteristics was conducted using Microsoft Excel 2019. A calculation of means, ranges, medians, standard deviations (SD) and maximum and minimum values was performed. All data reported in this study were analyzed by the first author (a junior health services researcher with a background in palliative nursing and experience in pain management) as part of a master’s thesis. All analytical steps and findings from both parts of the sample (GPs and patients) were considered and approved by a senior researcher (a research team member with profound expertise in qualitative research) during regular methods counselling sessions. In addition, the senior researcher reviewed all data and codes to ensure the validity of the findings. The general methodological approach and assigned codes were reflected repeatedly in research workshops with junior and senior researchers.

## 3. Results

A total of *n* = 21 GP and *n* = 40 patient interviews were conducted, which exceeded the expectation of the internal study protocol by far. In the patient sample, *n* = 12 individuals were using opioids at the time. For this study, all interviews containing relevant statements regarding the defined research objective (role models and expectations) were selected and a total of *n* = 25 interviews were analyzed (GP: *n* = 10, patient: *n* = 15, including *n* = 6 patients with opioid therapy). Categorized findings were linked to three relevant TDF domains: (1) *beliefs about capabilities* (referring to perceived competence, individual beliefs about capabilities, self-efficacy and self-confidence; patient expectations of themselves and GP expectations of patients were subsumed); (2) *beliefs about consequences* (comprising outcome expectancies and their perceived characteristics including beliefs and behavior that lead to outcomes; patient expectations of CNCP treatment and GP expectations referring to their assumptions of patient expectations were subsumed); and (3) *social/professional role and identity* (defined as “a coherent set of behaviors and displayed personal qualities of an individual in a social or work setting”, including professional boundaries and confidence; care coordinator and diagnostician, therapist and monitoring were subsumed) [32].

Included quotes were translated and pseudonymized (GP = general practitioner, P = patient, P_Opi = patient undergoing opioid therapy, Pos. = transcript position). All findings are presented descriptively along the categorized findings and linked to the relevant TDF domains (see Figure 1).

### 3.1. Participant Characteristics

All participants had sufficient proficiency in German to participate in the interviews and complete the accompanying sociodemographic questionnaire. Patients reported to suffer from CNCP of various etiologies: 60% (*n* = 9) mentioned musculoskeletal disorders (e.g., low-back pain, degenerative spondylolisthesis), 26.67% (*n* = 4) stated postinfectious syndromes (Lyme borreliosis or varicella zoster infections), and one patient suffered from migraines. A total of 66.67% of the patients (*n* = 10) frequently used NSAIDs, 53.33% (*n* = 8) used low or high potency opioids (e.g., Tilidine or Oxycodone) on a daily basis, 26.67% (*n* = 4) of the patients used neuropathic pain medication (e.g., Gabapentinoids) regularly, and two patients reported not taking any pain medication at all. Overall, 33.33% (*n* = 5) of the included patients were in specialized ambulatory pain treatment, while 26.67% (*n* = 4) utilized multimodal inpatient pain therapy. Only two patients were in pain-oriented psychotherapy and one patient participated in a self-help group. All GPs included in this study were fully trained and experienced. Some GPs obtained additional qualifications during their career: 50% (*n* = 5) were qualified in emergency medicine, 30% (*n* = 3) reported a qualification in palliative medicine, and 20% (*n* = 2) stated qualifications in specialized pain therapy, acupuncture and chiropractic. Table 1 details participant characteristics.

### 3.2. Beliefs About Capabilities

#### 3.2.1. Expectations of Patients Towards Themselves

Participating patients acknowledged their responsibility to contribute to diagnostics and therapy in a positive way. They stated that treatment could not be simply delegated to the GP but depended on their own initiative and information procurement. One patient mentioned expecting other patients to be brave and take risks with new treatment options. Other aspects mentioned referred to self-assessment of one’s health condition in daily life and a rational choice of own capabilities and abilities. One patient stated that they refuse to operate heavy machinery when feeling moderate or strong pain or impediments. Regarding therapy options, one participant felt that the responsibility for suitable decisions was with the patient. Most participants saw a need for personal initiative and therapy adherence, and expected these from other patients too.

*“I am always the main actor, it is up to me whether I get back on my feet or not […].”* (P22, Pos. 45)

#### 3.2.2. Expectations of GPs Towards Patients

GPs indicated that they expect a proactive stance regarding treatment participation from their patients. One GP explained that they expect patients to initiate contact proactively if there were any changes once the chosen therapy approach was successful. Another GP explained that they expect patients to actively prevent pain exacerbations. It was also mentioned that patients should acquire information and become experts on their own health condition. One GP stated that patients were more willing to cooperate in therapeutic interventions if they recognized that a physician invested time in their situation. Several described that changes in a pain situation were dependent on a patient’s will to persevere in therapy. In particular, active participation in physiotherapy was assumed to be an essential part of changing CNCP conditions. This expectation extended to cooperation in physiotherapy sessions, doing the exercises at home daily and information procurement about other physical exercises like yoga.

*“For example, he has to do his exercises for lumbar spine problems, we prescribe physiotherapy for him and then he doesn’t do his exercises and then he comes to me and says the pain isn’t getting any better.”* (GP21, Pos. 5)

GPs stated that they expect patients to report adverse effects of new medication as well as signs of overdose, particularly for opioid treatments. One GP explained that patients needed to watch out for themselves and were to contact the GP if unexpected effects occurred. Patients were expected to actively decide if they wanted to start a specific medication since they had to live with potential adverse effects. In the process of medication tapering, GPs expected patients to exactly follow their instructions. One GP stated that patients needed to initiate reduction or tapering because it would not be successful without their cooperation.

*“I can suggest it (reduction or tapering), but then there is usually no willingness on the other side. And then I know that it will not be carried out properly. […] but the patient has to initiate it and then it happens quite well.”* (GP06, Pos. 136)

### 3.3. Beliefs About Consequences

#### 3.3.1. Therapy-Related Expectations

##### Patient Perception

Patients stated that they expected their GPs to prescribe adequate medication to target their individual chronic pain condition. This was complemented by one participant mentioning the wish for an anesthetic epidural injection to reduce pain resulting from spinal canal stenosis. The migraine patient explained that the GP refused requests for triptan prescriptions due to a limited prescribing budget. Patients expected regular checking of their continuous medication regarding potential for reduction, tapering or change. They anticipated a transition to more contemporary and efficacious medications, as well as the application of pioneering therapeutics and medications, including cannabinoids. Some patients perceived that more comprehensive monitoring, including the assessment of pain intensity over time and blood levels associated with the underlying disease, as beneficial for their pain management.

Patients expected referrals to competent and specialized physiotherapists, specialized physicians and specialized pain therapists by the GPs. One patient expressed a desire for a recommendation to a nutritional expert because, in his opinion, nutrition influences his chronic pain condition as well as his underlying disease. Information transfer between GPs, specialized physicians and therapists was seen as crucial to effective treatment by patients. Yet patients felt a notable absence of such communication, which was seen as detrimental to their care. Therefore, they expected a more enhanced interdisciplinary communication among therapy stakeholders.

*“[…] exchange between doctors is still very little, for example, my psychotherapist does not exchange information at all with my GP. […] I think I would like to see more communication and perhaps a more coordinated approach, particularly when it comes to pain.”* (P14, Pos. 117–118)

Patients ascribed considerable significance to non-pharmacological treatment measures such as physiotherapy, therapeutic exercise, complementary medicine or dietary changes. They expected prescriptions for physiotherapy, water gymnastics or other therapy options depending on the underlying disease. However, they considered their GPs actual prescribing and individual counselling inadequate due to a lack of time and budget restrictions. One participant indicated an interest in detailed nutritional counseling yet expressed skepticism concerning the expertise of GPs in this field. Most of the patients stated that they expected their GP to inform them about treatment options more frequently, such as cannabinoids, acupuncture, relaxation techniques and complementary medical treatments.

*“My GP also does acupuncture. I’ve had that too, but I always have to initiate it. So, it doesn’t come from him.”* (P6, Pos. 50)

##### GP Perception

Anticipation of patient expectations concerning CNCP therapy in primary care varied among the GPs. One GP explained that patients expected active or interventional therapy approaches, others described patient aversions regarding mainly medication-based treatment approaches. In contrast, it was also stated that some patients expected analgesics prescriptions. GPs detailed that patients might focus on pharmacotherapy due to its convenient application and the assumption that medication implies recovery. Though GPs perceived a high patient demand for analgesics, they also indicated that patients were afraid of developing addiction and continuously increasing dosages, particularly regarding opioids. Several GPs stated that the reduction and tapering of analgesics were mainly suggested by patients not by GPs. After a first prescription of physiotherapy, GPs perceived those patients expected to obtain further prescriptions to intensify therapy. Cannabinoid-based medications were considered to be of high relevance for patients. GPs hypothesized patient education about adverse effects of certain medication may lead to a higher prevalence of actual adverse effects because patients would expect them to occur. One GP mentioned patients were not interested in information about psychological treatment aspects. Several GPs posited that patients would assume their GP believed their pain to be psychosomatic or not taken seriously when asked psychological questions. One GP indicated that patients desired an easy way to numb their pain with analgesics instead of changing their diet or doing therapeutic exercise. All GPs discussed the patient expectation to be completely free of pain after effective treatment. From the physicians’ perspective, this expectation was challenging, as they rather focused on reducing pain to a tolerable level, and a complete absence of pain was not realistic.

*“So, I do talk to people about how they unfortunately cannot expect to be completely free of pain, but that the goal is to get them to a level of pain that is as low as possible, but that they can still cope with in their everyday lives.”* (GP20, Pos. 70)

#### 3.3.2. Expectations of Emotional Support

##### Patient Perception

Considerable emphasis was placed on interpersonal aspects. Patients stated their expectation that GPs know a certain amount of their personal, sociocultural and health-related circumstances, to the extent that they could adequately plan the therapy with respect to their daily lives. Most patients expressed a desire for empathetic care, to be taken seriously, and for GPs to actively listen to their explanations and descriptions. The absence of these elements was attributed to a lack of motivation, time constraints and high patient volumes, considered to be resulting from health system issues requiring resolutions on a policy level. One participant stated that GPs were overwhelmed by having to establish a holistic treatment approach for CNCP. Most patients articulated an expectation or desire for GPs to invest more time in detailed discussions, history taking and emotional support.

*“So, in order to be able to support me, the first thing that would be necessary is that the people (GPs) actually have time to familiarize themselves with the subject matter […].”* (P12, Pos. 58)

##### GP Perception

GPs presumed that patients wanted them to invest time in discussions and history-taking. A common assumption among physicians was that patients expected them to demonstrate interest in their health conditions and to respect their individuality, particularly regarding shared decision-making. Strong emphasis was put on the perception that patients expect their concerns to be taken seriously and to be listened to.

*“[…] some patients may feel that they are not properly understood or taken seriously. And that is very important for patients.”* (GP03, Pos. 15)

#### 3.3.3. Expectations Regarding Patient Education

Patient education was seen as an essential aspect of CNCP treatment in general practice. Expectations associated with this subject showed considerable variation among the interviewed patients. Most of them mentioned that current patient education was insufficient and unsatisfactory. The majority of participants explained that they would expect more detailed education, information and explanations on treatment-related themes. The absence of such resources was attributed to constraints in time, competence and workload. Multiple themes emerged from the data, indicating that patients would appreciate receiving more information, e.g., biopsychosocial model of pain, nutrition, treatment options and interventions, psychological care, medical reports of specialized physicians, nursing care, underlying diseases and medication, as well as adverse effects.

*“[…] that perhaps patients should be informed in practice about the level of nursing care and what it is like to have a chronic illness. For example, I was not informed at all about how a chronic illness starts? How does a chronic illness affect everyday life?”* (P6, Pos. 204)

### 3.4. Social/Professional Role and Identity

#### 3.4.1. Care Coordinator and Diagnostician

For patients, GPs represented the initial point of contact for general CNCP management or pain exacerbations. One patient mentioned that the GP organized medication delivery by sending the prescription directly to the pharmacy chosen by the patient. Some explained that their GP was aware of their health-related and personal circumstances to the extent that they coordinated treatment accordingly. Exchange of medical reports and treatment coordination between involved medical specialists was seen an essential part of the GPs role. Patients attributed high relevance to receiving referrals to medical specialists from their GP.

*“[…] the GP is basically the one who bundles everything together, so all the in- formation comes together with him and he is the constant in my treatment. […] he is the one who always coordinates it and who refers me.”* (P14, Pos. 34)

GPs also saw coordinating interdisciplinary collaboration and information exchange as a highly relevant part of their role in CNCP treatment. They declared that they collaborate with several different medical specialists and physio- or psychotherapists. Some mentioned that initiating outpatient physio- or psychotherapy, inpatient rehabilitation or multimodal ambulatory or stationary pain therapy were among their important tasks in CNCP treatment. In instances where they felt their professional boundaries were exceeded, GPs mentioned that they refer patients to other qualified medical or therapeutic specialists for further evaluation, diagnostics and treatment.

*“So, I’m not so arrogant as to say that I’m the great pain therapist, then I refer them (to a pain therapist).”* (GP21, Pos. 115)

Some patients indicated that their GP assumed the role of diagnostician regarding their CNCP. One patient mentioned that the GP discussed the probability of a rheumatoid arthritis diagnosis and the potential for magnetic resonance imaging or additional diagnostic procedures including blood tests. In contrast, another participant articulated that his GP did not conduct CNCP diagnostics.

GPs felt responsible for gathering comprehensive information about the symptomatic complex a patient experienced. They indicated that the extent of their CNCP-related diagnostic processes might be decreased compared to medical specialists or pain therapists. Some emphasized that they conducted extensive verbal and physical examinations to ensure detailed diagnostics. One GP felt it was challenging to diagnose whether chronic pain was the main issue or addictive behavior in cases of high analgesics consumption. It was explained that the exclusion of malignant diseases was essential in the diagnostics of CNCP. Regarding diagnostic processes, one GP stated that it was very complex to evaluate organic disease etiologies as well as psychosomatic aspects. In the first history-taking appointment, GPs explained that they are responsible for gathering comprehensive information about the symptomatic complex the patient is experiencing.

*“What hurts, when does it hurt, what makes it better, what makes it worse? […] Also, what kind of treatments have you had before. Have you ever had an operation? What was the operation about? Did it get better as a result? Did you have an injection? […] What kind of medication has been used so far?”* (GP02, Pos. 21)

#### 3.4.2. Therapist

Patients felt GPs’ therapeutic role included discussing the results of diagnostic measures, offering treatment options, explaining pain etiologies, emotional support, and discussing the biopsychosocial model of pain and the psychological influence on pain intensity. Several patients articulated that their GPs offered and conducted non-medical treatment options such as acupuncture and osteopathic interventions, and provided information about non-pharmacological measures such as gymnastics and relaxation techniques. Patients saw the most relevant part of the therapeutic role of GPs to be prescribing physiotherapy and analgesics. In addition to information on intended and unintended medication effects, injections were considered to be a significant aspect of GP-led CNCP therapy. Some patients criticized the focus on injections and analgesics instead of verbal discussions of potential treatment options.

*“Discuss? I go in and say I have pain here and there. And then he says I can give you an injection. And after a few minutes I’m on my way again.”* (P_Opi5, Pos. 19)

GPs stated that patient education about the biopsychosocial model of pain, pathophysiology of underlying diseases and the psychological influences on CNCP condition were essential to their role as therapist. They mentioned that information should be individually tailored to different educational levels and personal circumstances. Regarding digital health applications, GPs articulated that they prescribe these based on patient age and capabilities. It was discussed that non-pharmacological measures played a highly relevant role in CNCP treatment. In addition to prescribing physio-, occupational and psychological therapies, GPs considered recommendations about nutrition, exercise, topical applications, weight reduction and relaxation techniques as an essential part of their therapeutic role in the context of CNCP treatment. The importance of psychological treatment in pain therapy was widely acknowledged and concerns were voiced about the availability of and access to psychotherapeutic options. GPs stated that they were responsible for maintaining the continuity of therapies initiated by medical specialists or hospital physicians and indicated that they reevaluated treatments started by other physicians, particularly regarding pharmacological measures. Pharmacological CNCP therapy was described as highly relevant. GPs emphasized that it included the selection of appropriate medication, dosage finding, medication rotation, exploration of potential applications, prevention of addiction, assessment of impairment of driving abilities, and information regarding the effects and adverse effects, as well as the reduction and tapering of medication, especially opioids. In particular, the selection of appropriate analgesics regarding comorbidities and adverse effects was considered challenging. With respect to treatment-related decisions, GPs perceived their role as advisory and stated that patients must decide between potential treatment options.

*“I advise and then tell the person in front of me that as an adult they can decide for themselves […]. I think that acceptance should be there.”* (GP06, Pos. 58)

#### 3.4.3. Monitoring

Patients felt the GPs role in CNCP management included monitoring the progression of pain over time, their psychological state and the status of underlying medical conditions, for instance through blood testing. They stated that GP responsibilities in the monitoring of CNCP encompasses the administration of medications, efficacy evaluation of applied treatment options, facilitation of brief consultations and the review of medical reports from other medical specialists engaged in monitoring activities. One patient mentioned that the GP used a systematic monitoring approach, partly including specific assessment instruments.

*“[…] then I get a questionnaire […] how was your pain this morning, this after- noon, this evening, this night? How did you feel? How did your pain affect your mood? What other side effects did you have?”* (P_Opi12, Pos. 84)

GPs characterized monitoring as an essential part of CNCP management in general practice, which included the number of prescriptions requested by patients, documentation and assessment of occurring adverse effects, respondence towards analgesics and other therapy options, but also pain development over time. In order to monitor adverse effects continuously, GPs included family members and ambulatory nursing services. One GP described monitoring a patient’s functionality to follow therapy progress. To assess the psychological well-being of patients, one GP employed a specialized depression scale on a regular basis and geriatric assessment options for older patients.

*“There is this depression scale that we use once a year, […] We also do these depression surveys for older people, i.e., Geriatric Basic Assessment.”* (GP05, Pos. 89)

## 4. Discussion

This study’s objective was to explore the expectations and perceived role models of GPs and patients regarding CNCP management in general practice. Our findings indicate that patients considered themselves as main actor in the therapy process yet expected guidance and care coordination from their GP. Although patient perspectives regarding the importance of the GP in CNCP management exhibited variability, the essential role GPs play in CNCP management was emphasized. GPs anticipated patients to put high relevance on pharmacological options. However, patients expressed a preference for non-pharmacological and alternative treatment options such as physiotherapy, acupuncture and exercise. Also, GPs expected their patients to actively engage in treatment and highlighted the importance of patient motivation to persevere with therapy. Contextualizing physiotherapy, GPs perceived that some patients were not adhering to their exercise routines. While GPs did not explicitly associate this noncompliance with a potential presence of kinesiophobia in CNCP patients, it might be a relevant consideration [35]. Participant perceptions of expectations and roles point to a strong connection between perceived role models and expectations as described by Dimbath et al. [33]. Patient expectations of GPs often referred to an ideal picture of how GP-led CNCP management should be. Role models indicated by GPs and some patients somehow represented a care ideal, which was also reflected in discussed expectations. Overall, this study provides important findings that will be incorporated into the development of the planned intervention.

Prior studies indicated a high relevance of individually tailored, holistic care approaches and the patient expectation that their care provider listens carefully and invests their time [18,19]. The findings of this study describe similar patient expectations in the context of German primary care. However, there are noticeable disparities in the perceptions of CNCP treatment between GPs and patients. GP role perceptions included expectations that were indicated by patients. This emphasizes the different perspectives on current practice and ideal care. A qualitative study from the UK stated that psychological therapy in the context of CNCP treatment is associated with social judgement and stigma [18]. The findings in this study hint at similar perceptions, though none of these assumptions were explicitly voiced in the conducted interviews. The majority of statements pertaining to psychological care were elicited from GPs during the course of the interviews, with only a few responses obtained from patients. Most of the patients linked the failure to meet their expectations to a lack of time and the high workloads of GPs. Many expectations described in this study are referred to in national guidelines on CNCP management [11,12]. In alignment with the patient expectations outlined in this study, the guideline on CNCP management published by the German Society of General Practice (DEGAM) states that non-pharmacological self-care interventions should be the primary and most prioritized treatment option. It places particular emphasis on physical activity as the most effective non-pharmacological treatment option [12]. In addition, the guidelines on the long-term use of opioids in CNCP highlight the importance of non-pharmacological interventions for patients with various underlying diseases [11]. The GPs who participated in this study did not address patients’ expectations regarding education or coping with the condition in daily life. The DEGAM guideline recommends structured history-taking, optionally supported by specific scales and assessments, and the importance of regular assessments for depression and anxiety [12]. It also points out potential existential concerns of patients affected by CNCP, underscores the importance of recognizing patients’ needs and providing advice on psychological care and treatment options, and the availability of care support centers. Such centers are explicitly designed to provide counsel on a range of topics, including nursing care and strategies for navigating daily life. They also offer guidance on insurance-covered options available for daily life support [12]. This strengthens the intervention development of RELIEF and demonstrates that the guidelines address patient needs. However, it also serves to highlight the potential for implementation barriers in the context of general practice. Contemplated roles and expectations of GPs and patients imply that a structured, guideline-based case and care management program potentially could positively influence care quality and provider and patient satisfaction. GPs as well as patients described the relevance of referrals to medical specialists, especially specialized pain therapists. In this regard, it should be noted that in some areas in Germany, few specialized pain services are available.

Most patients participating in this study lived in a partnership or had children. Contemporary demographic trends suggest an increase in the proportion of childless individuals for the future [36]. The future of CNCP management will therefore need to address additional challenges resulting from an increased number of people living alone or without relatives. Patients saw information procurement as part of their role and attributed educational aspects mainly to GP care, neglecting options offered by self-help groups or insurance providers. This indicates a need for the low-threshold dissemination of corresponding information in general practices, which could support a sustainable use of GP resources. The exploration of adequate informative formats thus appears to be recommendable.

### Strengths and Limitations

Individual interviews and structuring qualitative content analysis proved to be a suitable approach to explore expectations and perceived role models of GPs and patients. Interviewers flexibly used the interview guides to enable establishing pertinent points and to maintain rapport. Including both the GP and patient perspectives supported the exploration of various perceptions of CNCP management in general practice. This can be considered a significant strength of this study. All analytical steps and findings from both parts of the sample were reflected on and consented to by a senior qualitative researcher to ensure the validity of the findings. Also, the general methodological approach and assigned codes were reflected on repeatedly in research workshops with junior and senior researchers. The patient sample exhibited a multitude of etiologies of CNCP, heterogeneous ages and occupational profiles. Since participants were recruited from various sources, it can be assumed that the sample reflects urban and rural population and care settings for general practice.

The TDF was applied as a methodologically established framework to contextualize and classify behavior perceptions. Most concepts in TDF were not relevant to this study. Despite definitional advantages, the TDF domains only partially reflected subcategories. Consequently, the main categories derived from the TDF were subjected to further differentiation at the subcategory level, extending beyond the TDF. Data analysis was conducted by the first author who was not involved in data collection. This facilitated a high degree of objectivity and independence, which enhances the rigor of this study. All transcripts containing relevant statements (*n* = 25) were included in the analysis even though no further themes emerged from the data after the analysis of *n* = 18 transcripts (*n* = 8 GPs; *n* = 10 patients). Contextualizing the results of this study with the international literature [37] thus indicates thematic saturation for both interviewed groups. However, additional data collection may yield novel insights. There was no correction of transcripts or feedback on findings from participants or repeated interviews. The reporting of this study follows the Consolidated Criteria for Reporting Qualitative Research (COREQ) [38].

Some limitations have to be reported. Since a random selection of individuals were contacted for participation, it was unknown whether they suffered from CNCP. Contacting individuals currently receiving CNCP treatment may result in a higher participation rate and more comprehensive data on the subject. Patient perspectives on GP expectations were less detailed than GP expectations of patients. Expanding on patient expectations might thus provide a more balanced view. Interviewers may have missed opportunities for prompts to evoke in-depth narrative. Social desirability regarding verbalized statements cannot be excluded. There is also a risk of interviewer bias. It is possible that interviewers’ preconceived notions or focus on specific aspects resulted in an overrepresentation of data pertaining to such aspects. Several GPs had obtained additional qualifications, such as palliative care or specialized pain therapy, which may have resulted in an increased level of knowledge and competence regarding CNCP treatment. The first author is a junior researcher in health services research with a background in palliative nursing care. Preconceptions stemming from this background could potentially have influenced data analysis.

## 5. Conclusions

GP and patient perceptions on role models and expectations highlight the importance of a guideline-based holistic, individually tailored CNCP management in general practice. Systematic, multifaceted case management in general practice such as planned in the RELIEF project may prove essential to ensuring the delivery of high-quality primary care for individuals with CNCP. A strong emphasis on non-pharmacological interventions, interdisciplinary and interprofessional collaboration, and specialized pain treatment in clinical practice and research holds promise for enhancing patient pain management.

## Figures and Tables

**Figure 1 healthcare-13-00187-f001:**
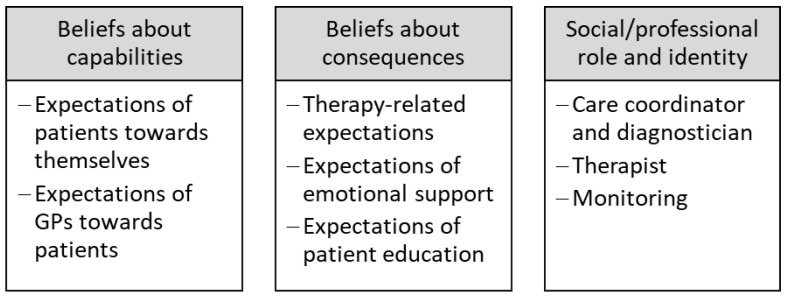
Theoretical Domains Framework domains and linked subcategories.

**Table 1 healthcare-13-00187-t001:** Participant characteristics (*n* = 25).

	GPs (*n* = 10)	Patients (*n* = 15)
Age: mean (SD), range	50 (12.72), 32–71	69.6 (13.04), 40–84
Gender: *n* (%) male	4 (40%)	5 (33.33%)
Experience in years: mean (SD), range	20.6 (11.45), 4–42	
Marital status: ^a^ *n* (%) in partnership		10 (66.67%)
Housing situation: *n* (%) living alone		5 (33.33%)
Employment status: *n* (%) retired		10 (66.67%)
Children: *n* (%) with ≥1 child		14 (93.33%)
Pain duration in years: mean (SD), range		19 (17.19), 2–50
Pain intensity ^b^ in the last 4 weeks: mean (SD), range		5.53 (2.32), 2–9

^a^ binary variable: in partnership or not; ^b^ measured with numeric rating scale for pain assessment (0–10) [34].

## Data Availability

Data generated and analyzed in this study are kept on secure servers at the Department of Primary Care and Health Services Research, University Hospital Heidelberg, Germany. On reasonable request, de-identified datasets can be made available by the corresponding author.

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
