# Peer review of "Exploring Physician and Patient Perspectives on Expectations and Role Models Towards Chronic Pain Treatment in General Practice: A Qualitative Cross-Sectional Study"

_healthcare, 2025, doi:10.3390/healthcare13020187_

Round 1
Reviewer 1 Report
Comments and Suggestions for Authors
Thank you for a nice study exploring chronic pain management in primary health setting. The methods are well described, although I can not fully understand the selection process of the participants. A purposive sampling strategy was used, but it is not described what the aim was (the purpose). Was the intention to get a spread in age, gender, living conditions, income, rural/large cities, or what? This is obscure in the manuscript and needs to be described. In current form it only states how many participants where invited but we do not know whether the selection was based on any other criterias.
Another thing I miss from the methods is the researchers pre-knowledge. In a qualitative study the researcher use their own pre-kowledge actively during the research and therefor it is important to be aware of your own preknowledge and describe it in the article.
Another thing I find missing in the description is how saturation was established and when. In the description of the study population the authors state that "a sample size of 40 participants was considered appropriate to reach thematic saturation" which often is true, but it needs assessment in order to be true. Often during the analysis one finds that saturation might be reached after e.g. 35 participants and when another interview is analyzed no new themes appear so then you perform a couple of more analyses in order to confirm this.
In this study there are two groups of participants, GP's and CNCP-patients. Therefore a thematic saturation ought to be reached in each group. How this was confirmed needs to be described.
Lastly, how was the results triangulated in order to make sure it was valid and reliable? Was it triangulated against the groups of participants? Against another group or how?
The results follow the aim of the study and not much to say about this. One of the strengths of the study is the two perspectives, the patients and the GP's. This could be emphasized better.
In the discussion it gets apparent that one of the main pre-knowledge (prejudice) is the interpretation of the guidelines that emphasize a holistic treatment approach. The GP's describe that the patients are prescribed physiotherapy and then do not perform the exercises and therefore are responsible for their lack of improvement. But no question is raised why patients don't perform the exercises despite several studies showing that physiotherapy and exercise often makes pain worse and only seldom reduce pain. A discussion with a more open mind would have been appreciated and might have revealed new knowledge in the field!
Despite my concerns, I think this study is well written and interesting. Please add the missing parts of the methods section, and improve the discussion in order to be a bit more open minded. If possible, triangulate the results by either short telephone interviews with the participants or a small focus group of GP's.
Author Response
Thank you for a nice study exploring chronic pain management in primary health setting. The methods are well described, although I can not fully understand the selection process of the participants. A purposive sampling strategy was used, but it is not described what the aim was (the purpose). Was the intention to get a spread in age, gender, living conditions, income, rural/large cities, or what? This is obscure in the manuscript and needs to be described. In current form it only states how many participants where invited but we do not know whether the selection was based on any other criteria.
Dear Reviewer 1,
thank you very much for taking the time to review our manuscript and for your recommendations. Adjustments regarding your proposals were added in line 140-143.
Another thing I miss from the methods is the researchers pre-knowledge. In a qualitative study the researcher use their own pre-kowledge actively during the research and therefor it is important to be aware of your own preknowledge and describe it in the article.
Thank you for drawing our attention to this aspect. As stated in section 2.1, the RELIEF project's initial phase entailed a comprehensive literature review conducted by the project team. This review underscores the team's extant knowledge. We have adapted the text now (lines 105 to 107) to describe that the knowledge derived from the literature review was incorporated in the subsequent research.
Another thing I find missing in the description is how saturation was established and when. In the description of the study population the authors state that "a sample size of 40 participants was considered appropriate to reach thematic saturation" which often is true, but it needs assessment in order to be true. Often during the analysis one finds that saturation might be reached after e.g. 35 participants and when another interview is analyzed no new themes appear so then you perform a couple of more analyses in order to confirm this.
Thank you for your attention to this important issue. In the "Strengths and Limitations" section, we have included the number of interviews analyzed after which no new themes emerged from the data. This information can be found on lines 621 to 626.
In this study there are two groups of participants, GP's and CNCP-patients. Therefore, a thematic saturation ought to be reached in each group. How this was confirmed needs to be described.
We appreciate your attention to this issue. In line 625, we revised our text passages to enhance transparency regarding thematic saturation.
Lastly, how was the results triangulated in order to make sure it was valid and reliable? Was it triangulated against the groups of participants? Against another group or how?
Thank you for pointing these important aspects out. The project team engaged in a thorough analysis and interpretation of the data from both groups to ensure the integrity and coherence of the results. Nevertheless, we added the missing information now to the Strengths and limitations section.
The results follow the aim of the study and not much to say about this. One of the strengths of the study is the two perspectives, the patients and the GP's. This could be emphasized better.
Thank you very much. We added a brief sentence in lines 609-610 to underline this strength of the study.
In the discussion it gets apparent that one of the main pre-knowledge (prejudice) is the interpretation of the guidelines that emphasize a holistic treatment approach. The GP's describe that the patients are prescribed physiotherapy and then do not perform the exercises and therefore are responsible for their lack of improvement. But no question is raised why patients don't perform the exercises despite several studies showing that physiotherapy and exercise often makes pain worse and only seldom reduce pain. A discussion with a more open mind would have been appreciated and might have revealed new knowledge in the field!
Thank you for this remark, in lines 542 to 545 we added this consideration supported by literature.
Despite my concerns, I think this study is well written and interesting. Please add the missing parts of the methods section, and improve the discussion in order to be a bit more open minded. If possible, triangulate the results by either short telephone interviews with the participants or a small focus group of GP's.
Reviewer 2 Report
Comments and Suggestions for Authors
1. Title The article's title is clear and adequately reflects the study's content. However, it could be simplified for conciseness and to better capture readers' attention. For example, "Physicians' and Patients' Perspectives on Chronic Pain Treatment in Primary Care: A Qualitative Study" might be a more direct option.
2. Abstract The abstract provides an adequate overview of the study, including objectives, methods, results, and conclusions. However, the information could be better organized to highlight key findings and the clinical relevance of the study. It is also recommended to avoid excessive technical jargon to improve accessibility to a broader audience.
3. Introduction The introduction appropriately contextualizes the importance of chronic pain management in primary care and clearly outlines the study's objectives. However, it could be enriched with a broader discussion on the existing gap between physicians' and patients' expectations in chronic pain management.
4. Methods The study design is well described, and the use of semi-structured interviews is appropriate for exploring qualitative perspectives. The choice of the Theoretical Domains Framework (TDF) as the analytical framework is suitable. However, the description of sampling and the justification for sample size could be more detailed to strengthen the study's validity.
5. Results The results are well organized and provide a comprehensive view of physicians' and patients' expectations. Nevertheless, it is recommended to include more direct quotes to better illustrate the qualitative findings. Additionally, a deeper discussion on the discrepancies between physicians' and patients' perspectives could enhance the analysis.
6. Discussion The discussion contextualizes the findings within the existing literature and underscores the importance of holistic, personalized chronic pain management. However, it could benefit from a more critical reflection on the study's limitations and practical implications for primary care. Furthermore, a deeper exploration of the barriers to implementing the study's recommendations in daily clinical practice is warranted.
7. Conclusions The conclusions are well-founded and reflect the study's findings. However, it would be helpful to provide more specific recommendations for clinical practice and suggestions for future research in this area.
8. References The references are relevant and current, but the literature review could be expanded to include additional studies examining multifaceted interventions in chronic pain management in primary care.
Final Recommendation The study addresses an important topic and provides useful findings for clinical practice. Nonetheless, minor revisions are recommended to improve the clarity and depth of the analysis. With these adjustments, the article would be suitable for publication in the journal.
Author Response
- TitleThe article's title is clear and adequately reflects the study's content. However, it could be simplified for conciseness and to better capture readers' attention. For example, "Physicians' and Patients' Perspectives on Chronic Pain Treatment in Primary Care: A Qualitative Study" might be a more direct option.
Dear Reviewer 2,
thank you very much for your recommendations and for taking the time to assess our manuscript. We thoroughly considered your suggestion regarding the title, however, we decided not to change it as we felt that simplifying the title would not reflect the aim of our study completely.
- AbstractThe abstract provides an adequate overview of the study, including objectives, methods, results, and conclusions. However, the information could be better organized to highlight key findings and the clinical relevance of the study. It is also recommended to avoid excessive technical jargon to improve accessibility to a broader audience.
Thank you for this remark. We thoroughly re-checked the abstract regarding technical jargon and concluded that we don’t see opportunity for improvements. With regard to key findings, we briefly added some information.
- IntroductionThe introduction appropriately contextualizes the importance of chronic pain management in primary care and clearly outlines the study's objectives. However, it could be enriched with a broader discussion on the existing gap between physicians' and patients' expectations in chronic pain management.
Thank you for pointing out this important aspect. Currently, the discrepancy between physicians' and patients' expectations referring to CNCP management in primary care in Germany is not yet well documented in existing literature which is why we aimed to contribute knowledge to the field regarding these aspects. Further studies in the RELIEF project (and in further studies) will add to the findings reported in this manuscript. We added some text regarding this aspect to the introduction now.
- MethodsThe study design is well described, and the use of semi-structured interviews is appropriate for exploring qualitative perspectives. The choice of the Theoretical Domains Framework (TDF) as the analytical framework is suitable. However, the description of sampling and the justification for sample size could be more detailed to strengthen the study's validity.
Thank you very much for this recommendation. One other reviewer also pointed this out to us and we made the necessary amendments.
- ResultsThe results are well organized and provide a comprehensive view of physicians' and patients' expectations. Nevertheless, it is recommended to include more direct quotes to better illustrate the qualitative findings. Additionally, a deeper discussion on the discrepancies between physicians' and patients' perspectives could enhance the analysis.
Thank you very much, we modified the discussion as well as the limitations and strengths section in order to refine the discursive aspects.
- DiscussionThe discussion contextualizes the findings within the existing literature and underscores the importance of holistic, personalized chronic pain management. However, it could benefit from a more critical reflection on the study's limitations and practical implications for primary care. Furthermore, a deeper exploration of the barriers to implementing the study's recommendations in daily clinical practice is warranted.
Thank you for mentioning these very valid aspects. Regarding the study’s limitations we added various points in lines 629-634 as well as 636-638.
- ConclusionsThe conclusions are well-founded and reflect the study's findings. However, it would be helpful to provide more specific recommendations for clinical practice and suggestions for future research in this area.
Thank you for your recommendation, we added some suggestions for research as well as clinical practice in line 650-653.
- ReferencesThe references are relevant and current, but the literature review could be expanded to include additional studies examining multifaceted interventions in chronic pain management in primary care.
As described in the introduction, only a limited amount of studies investigated multifaceted interventions on CNCP in the primary care setting. We referenced them briefly.
Reviewer 3 Report
Comments and Suggestions for Authors
Dear Authors,
Thank you for providing me with the opportunity to read thsi interesting piece of work. Below, I have listed my comments:
Title/Abstract:
1) The phrase 'role models'is a little unclear and mighe be unclear to readers too. Does it refer to ideal behaviors, best practices, or specific individuals? A more precise term could improve clarity.
Introduction:
1) In line 41, data from 2014 feels outdated without justification. If it is not possible to identify newer statistics, you could explain why 2014 data is still relevant or highlight the persistence of the issue over time.
2) The introduction mentions the lack of German context and comprehensive interventions toward the end. To better frame the study's rationale, it might be better to highlight the gaps throughout the introduction.
3) The objective of the study is stated but it would have been nice to also emphasise the important or some brief implications of the findings. Also, there could have been a brief connection of the CNCP management to its global significance or relevance in other healthcare systems to make the introduction more engaging for international audiences.
Methods
1) The sampling needs a stronger justification. Why 40 participants were recruited? What would happen if more individuals were willing to participate? For instance, explain how it aligns with thematic saturation for qualitative studies and cite supporting literature.
2) The piloting process for interview guides is mentioned briefly. Provide a bit more detail about what was learned from piloting (e.g., specific adjustments made to improve coherence or question phrasing).
3) Were there any potential biases or limitations of the recruitment strategy? Need to acknowledge them.
4) While ethical compliance is addressed, it could be expanded to mention participant confidentiality, data security measures, and how informed consent was ensured for both in-person and telephone interviews.
Results
1) There is a point that is confusing. In the methods, it is stated that 40 participants were considered appropriate but in the results it is stated that 'A total of n=21 GP and n=40 patient interviews were conducted.' meaning 61 interviews and then it is stated that 'a total of n=25 interviews were analyzed'. Could you please clarify how many individuals participated in this study?
2) why only 25 of the 61 interviews were selected for analysis? This could create potential concerns about selection bias. How was this addressed?
3) The GP expectations toward patients (e.g., adherence to therapy, medication management) are explored extensively, but patient perspectives on GP expectations are less detailed. Expanding the latter would provide a more balanced view.
Discussion
1) While the role of GPs is well-explored, the discussion could delve deeper into patients' responsibility for self-care and how to empower them in a structured manner.
2) Some points, such as interviewer bias and preconceptions, are mentioned but not fully explained. For instance, how did these biases potentially shape the results?
3) The mention of stigma around psychological care (lines 545–547) is underdeveloped. Exploring how this affects CNCP management in Germany (or comparing with other countries) could strengthen this point.
References
The numbers are doumpled.
I hope this feedback is helpful.
Author Response
Title/Abstract:
1) The phrase 'role models' is a little unclear and mighe be unclear to readers too. Does it refer to ideal behaviors, best practices, or specific individuals? A more precise term could improve clarity.
Dear Reviewer 3,
thank you very much for assessing our manuscript and your recommendations. Since we share your concern regarding the terms role models and expectations, we had included a literature-based sociological definition in the text already (lines 218-224) to provide clarity.
Introduction:
1) In line 41, data from 2014 feels outdated without justification. If it is not possible to identify newer statistics, you could explain why 2014 data is still relevant or highlight the persistence of the issue over time.
Thank you for this remark, we underlined the persistent relevance of the issue in lines 46-47.
2) The introduction mentions the lack of German context and comprehensive interventions toward the end. To better frame the study's rationale, it might be better to highlight the gaps throughout the introduction.
Thank you for your remark. We absolutely agree with you and added some aspects regarding international comparisons on the utilization of specialized pain treatment, line 70-71 and regarding the knowledge gap referring to expectations of GPs and patients.
3) The objective of the study is stated but it would have been nice to also emphasise the important or some brief implications of the findings. Also, there could have been a brief connection of the CNCP management to its global significance or relevance in other healthcare systems to make the introduction more engaging for international audiences.
Thank you very much for pointing this out to us. We now added an international prevalence estimation in order to highlight the global significance of the disease (line 41-45). Regarding international utilization of specialized pain management, we also added some contextualizing data in line 70-71.
Methods
1) The sampling needs a stronger justification. Why 40 participants were recruited? What would happen if more individuals were willing to participate? For instance, explain how it aligns with thematic saturation for qualitative studies and cite supporting literature.
Apologies for the omission. We have added information regarding saturation now to the strengths and limitation section (line 621-625). Supporting literature had already been cited.
2) The piloting process for interview guides is mentioned briefly. Provide a bit more detail about what was learned from piloting (e.g., specific adjustments made to improve coherence or question phrasing).
Thank you for pointing us to this aspect. In line 175-177 we specified the adaptations of the patient interview guide briefly. Terminology was clarified to increase clarity of the questions.
3) Were there any potential biases or limitations of the recruitment strategy? Need to acknowledge them.
Thank you very much for this remark, we thoroughly documented the recruitment process and after revisiting the documentation and discussion in the team, we added some aspects in the limitations part of the manuscript now, particularly in line 629-634.
4) While ethical compliance is addressed, it could be expanded to mention participant confidentiality, data security measures, and how informed consent was ensured for both in-person and telephone interviews.
Thank you for mentioning this aspect. In line 161 we specify that after expressing interest in participating in an interview, patients were sent an informed consent form. Additionally, an informed consent discussion was conducted via telephone (line 163). The aspect of participant confidentiality is added now in the text (line 162-163).
Results
1) There is a point that is confusing. In the methods, it is stated that 40 participants were considered appropriate but in the results it is stated that 'A total of n=21 GP and n=40 patient interviews were conducted.' meaning 61 interviews and then it is stated that 'a total of n=25 interviews were analyzed'. Could you please clarify how many individuals participated in this study?
Apologies for any confusion here. Indeed, based on researcher’s prior experiences, the internal study protocol considered it appropriate to conduct a total of 40 interviews. However, patients were very eager to participate and thus we could recruit 40 patients (out of 50 who signaled interest in participation). Since we did not want to reject eligible participants, we decided to conduct interviews with all eligible individuals who could be scheduled within the given timeframe. This was not possible for all interested individuals. Thus, we could conduct a total of 61 interviews. Not all interviews contained statements regarding roles and expectations, therefore the 25 transcripts that documented such statements referring to the research objective were analyzed in this study. We have slightly added to the Results section to increase transparency (line 235-239).
2) why only 25 of the 61 interviews were selected for analysis? This could create potential concerns about selection bias. How was this addressed?
As mentioned above and in the manuscript text, all interviews containing statements regarding roles and expectations were included in the analysis. We included a brief pointer now at the beginning of the Results section to increase transparency.
3) The GP expectations toward patients (e.g., adherence to therapy, medication management) are explored extensively, but patient perspectives on GP expectations are less detailed. Expanding the latter would provide a more balanced view.
Thank you very much for mentioning this. We absolutely agree with you regarding this aspect. However, since we cannot expand on it based on our data, we added a recommendation for further research to explore this aspect in more details now to the manuscript in the Limitations section (line 632-634).
Discussion
1) While the role of GPs is well-explored, the discussion could delve deeper into patients' responsibility for self-care and how to empower them in a structured manner.
Thank you also for pointing this out. We had discussed this idea previously in the research team but decided not to include it in this manuscript as it will be part of another manuscript.
2) Some points, such as interviewer bias and preconceptions, are mentioned but not fully explained. For instance, how did these biases potentially shape the results?
Thank you for your remark, in line 636-638 we added how interviewer bias or preconceptions may have influenced the results.
3) The mention of stigma around psychological care (lines 545–547) is underdeveloped. Exploring how this affects CNCP management in Germany (or comparing with other countries) could strengthen this point.
Thank you very much, in our opinion this is a highly relevant aspect. Nevertheless, psychological care and stigma were not addressed explicitly in the patient interviews. After discussing the issue with the research team, we decided against focusing on this aspect because it is not emphasized in our data.
References
The numbers are doumpled.
Thank you very much for detecting the doubled numbers. This seems to have occurred during the transfer of the manuscript onto the provided template and went unnoticed. We corrected the reference list accordingly now.
I hope this feedback is helpful.
Thank you very much for your kind and constructive feedback, it is highly appreciated.
Round 2
Reviewer 1 Report
Comments and Suggestions for Authors
I still think the manuscript would be improved by a more thorough description of the researchers pre-knowledge. Now it seams like they where totally new in primary health and pain management and their only pre-knowledge was based on the findings in the literature study in the initial phase. If that is the case, there is a risk that a lot of information was lost during interview and analysis. If at least one of the researchers had pre-kowledge of working in primary healthcare and/or working with chronic pain patients this risk is reduced significantly. Triangulation of the results are not described either, and if there is no pre-knowledge among the researchers a triangulation against GP's and patients is of uttermost importance in order to make sure the results are valid. especially since the authors state in the introduction that there are a lack of published studies (line 87-88) from Germany and that the population of pain patients in Germany differ from other parts of the world in how they search help (Line 70-71).
Author Response
I still think the manuscript would be improved by a more thorough description of the researchers pre-knowledge. Now it seams like they where totally new in primary health and pain management and their only pre-knowledge was based on the findings in the literature study in the initial phase. If that is the case, there is a risk that a lot of information was lost during interview and analysis. If at least one of the researchers had pre-kowledge of working in primary healthcare and/or working with chronic pain patients this risk is reduced significantly.
Thank you Reviewer 1 for again taking time to go through our manuscript. We understand your concern regarding pre-knowledge. However, we would like to point out that among our group of authors are two very experienced General practitioners who besides actively practicing medicine also engage in research and in the development of primary care guidelines referring to chronic pain (PD Dr. med. Cornelia Straßner, PD Dr. med. Peter Engeser), an internationally renowned professor in Health Services Research and Implementation Science with decades of expertise in research in primary care (Prof. Dr. Michel Wensing), a senior researcher in Health Services Research and Implementation Science with a number of years of experience in primary care research (Dr. sc.hum. Regina Poß-Doering), and three junior Health Services Researchers with backgrounds in pediatric and palliative nursing care. All authors with a medical professional background have practical experience in pain management. We therefore cannot agree with you on the notion that pre-knowledge was based on the literature study only. To increase transparency here, we have added some information to the text now and hope your concern has been dispelled.
Triangulation of the results are not described either, and if there is no pre-knowledge among the researchers a triangulation against GP's and patients is of uttermost importance in order to make sure the results are valid. especially since the authors state in the introduction that there are a lack of published studies (line 87-88) from Germany and that the population of pain patients in Germany differ from other parts of the world in how they search help (Line 70-71).
Thank you for addressing this aspect. We understand triangulation as refering to the use of multiple methods or data sources to facilitate a comprehensive understanding of phenomena (see Patton, 1999). In qualitative research, triangulation has been considered as a strategy to test validity by converging information from different sources. Four types can be considered: (1) method triangulation, (2) investigator triangulation, (3) theory triangulation, and (4) data source triangulation (see Denzin 1978 and Patton 1999). Though we do not explicitly use the word “triangulation” in our manuscript, we analyzed data from two target groups, i.e. GPs and patients (data source triangulation) in our study. Analysis and all findings have been reflected and consented with a senior researcher in the research team during methods counselling sessions. In addition, the senior researcher went over all data and codes to ensure findings are well grounded and valid (investigator triangulation). In addition, the general methodological approach and the assigned codes were reflected repeatedly in research workshops with junior and senior researchers. To increase transparency regarding the triangulation, we have added to the already provided information in the Methods section and under 'Strengths and limitations' now and hope this will dispell your concern.
Reviewer 3 Report
Comments and Suggestions for Authors
Dear Authors,
Thank you for responding to the feedback and for making the revisions. I am happy with the changes made. Good luck with the publication.
Author Response
Thank you for responding to the feedback and for making the revisions. I am happy with the changes made. Good luck with the publication.
Thank you for again taking the time to assess our manuscript and your kind comments.